# Piezoelectric and Dielectric Properties in Bi_0.5_(Na,K)_0.5_TiO_3_-x Ag_2_O Lead-Free Piezoceramics

**DOI:** 10.3390/ma16155342

**Published:** 2023-07-29

**Authors:** Xiaoming Chen, Yunwen Liao

**Affiliations:** 1School of Materials and Energy Engineering, Guizhou Institute of Technology, Guiyang 550003, China; 2College of Chemistry and Engineering, China West Normal University, Nanchong 637009, China

**Keywords:** lead-free, ceramics, phase transition, relaxor behavior

## Abstract

Lead-free piezoceramics of Bi_0.5_(Na_0.825_K_0.175_)_0.5_TiO_3_ with varying concentrations of x mol% Ag_2_O (x = 0, 0.1, 0.3, 0.5, 0.7, 0.9, denoted as BNKT-xA) were fabricated using the solid-state technique. An extensive investigation was undertaken to analyze the structural, piezoelectric, and dielectric properties of these piezoceramics in the presence of Ag ions. There is no evidence of any secondary solid solution in the BNKT-xA piezoceramics. The ceramics with x mol% Ag_2_O in BNKT still demonstrate the presence of both rhombohedral (R) and tetragonal (T) phases. The addition of Ag^+^ is helpful to increase the relative density of the BNKT-xA piezoceramics. It is noteworthy that the BNKT-0.3 mol% A piezoceramics show remarkable improvements in their properties (*d*_33_ = 147 pC/N, *k_p_* = 29.6%, *ε* = 1199, tan*δ* = 0.063). These improvements may be ascribed to the denser microstructure and the preservation of morphotropic phase boundaries between R and T phases caused by the appropriate addition of Ag cations. The addition of Ag^+^ results in the relaxor behavior of the BNKT-xA ceramics, characterized by disorder among A-site cations. With the increase in temperature, the *d*_33_ value of BNKT-xA ceramics does not vary significantly in the range of 25 to 125 °C, ranging from 127 to 147 pC/N (with a change of *d_33_* ≤ 13.6%). This finding shows that piezoelectric ceramics have a reliable performance over a certain operating temperature range.

## 1. Introduction

Piezoceramics are extensively employed in various applications, including transducers, sensors, actuators, etc. However, it is worth noting that the majority of piezoelectric ceramics used are Pb-based, specifically Pb(Zr_1−x_Ti_x_)O_3_ (PZT). The presence of Pb in these ceramics raises concerns regarding environmental toxicity, particularly attributed to the volatilization of PbO in the sintering procedure. Consequently, it has resulted in the implementation of legislation in different countries, such as regulations on WEEE and restrictions on RoHS [1,2]. Hence, it is important to focus on the development of lead-free piezoceramics from an environmental perspective.

Lead-free piezoceramics should be classified into four main species: BaTiO_3_-based [3,4], Bi_0.5_Na_0.5_TiO_3_-based, Bi-layered [5], and niobate-based ceramics [6]. Among these, Na_0.5_Bi_0.5_TiO_3_ (BNT) lead-free ceramics were first synthesized by Smolenkii et al. in 1960 and exhibit a rhombohedral crystal phase around room temperature, with a *T_c_* of 320 °C [7]. BNT ceramics display strong ferroelectricity, excellent piezoelectric performance, outstanding acoustic properties, and a lower sintering temperature compared to other systems [8,9]. As a result, BNT has become one of the most extensively studied Pb-free ceramics. However, pure BNT ceramics face challenges in practical applications due to their high coercive field and poor compactness. Currently, research on BNT-based ceramics primarily focuses on enhancing their performance through replacement and doping strategies. For instance, studies have been conducted on BNT-BaTiO_3_ [10,11], Bi_0.5_(Na_1−x−y_Li_y_K_x_)_0.5_TiO_3_ [12], Bi_0.5_Na_0.5_TiO_3_-K_0.5_Bi_0.5_TiO_3_ (BNKT) [13], Bi_0.5_Na_0.5_TiO_3_-KNbO_3_ [14], Bi_0.5_Na_0.5_TiO_3_-BaTiO_3_-KNbO_3_ [15], [(K_0.175_Na_0.825_)_0.5_Bi_0.5_]_1+x_TiO_3_ [16], (Na_1/2_Bi_1/2_)TiO_3_-BaTiO_3_-(Na_1/2_K_1/2_)NbO_3_ [17], and 0.76Bi_0.5_Na_0.5_TiO_3_-Bi(Ni_2/3_Nb_1/3_)O_3_-0.24SrTiO_3_ [18]. Among these materials, BNKT ceramics, particularly those near the morphotropic phase boundary (MPB) between the rhombohedral and tetragonal phases with 0.16 ≤ x ≤ 0.2, exhibit excellent electrical properties, such as high ratios of *k_t_* (the piezoelectric coupling coefficient) to *k_p_* (the planar coupling coefficient). Furthermore, the incorporation of Bi_0.5_K_0.5_TiO_3_ into BNT can reduce the coercive field (*E_c_*) to a minimum of 4 kV/mm [13]. Nevertheless, the high sintering temperature required for this material can lead to the evaporation of Na and K [19].

Grinberg and Rappe [20] reported that the presence of Ag in perovskite solid solutions leads to significant polarization and piezoelectric responses due to the large distortion of Ag on the A-site, as observed in the AgNbO_3_ system [21]. Wu et al. discovered that when Bi_0.5_Ag_0.5_TiO_3_ is introduced into BNT-BT, the piezoceramics are still located at the R–T phase transformation. Additionally, these ceramics not only achieve dense structure but also have optimum electric performance (*d*_33_ = 172 pC/N, *k_p_* = 32.6%, *k_t_* = 52.6%) [22]. Lin et al. observed a morphotropic phase boundary (MPB) in (Li_0.075_Na_0.925−x−y_Ag_y_K_x_)_0.5_Bi_0.5_TiO_3_ systems where K^+^, Li^+^ and Ag^+^ occupy part of the BNT lattices. Specifically, in the system with 0.15 < x < 0.25, the presence of the morphotropic phase boundary, combined with a low coercive field, significantly enhances piezoelectric properties (*d*_33_ = 178–219 pC/N, *k_p_* = 35–39%, *k_t_* = 44–51%) [23]. Liao et al. found that the substitution of Ag^+^ and K^+^ for Na ions in BNT results in the high performance of (Na_1−x−y_Ag_y_K_x_)_0.5_ Bi_0.5_TiO_3_ piezoceramics (*d*_33_ = 180 pC/N, *k_p_* = 0.35) [24]. These studies suggest that the combination of the presence of Ag and phase boundary effects holds significant potential for the manufacture of BNKT-based ceramics with improved performance.

In this case, an appropriate amount of Ag_2_O was introduced into BNT-Bi_0.5_K_0.5_TiO_3_ to enhance its sinterability and performance. The traditional ceramic process is employed to prepare the piezoceramics. The structure and electrical properties are systematically studied. A comprehensive elucidation of the underlying mechanism is presented in detail.

## 2. Materials and Methods

According to the traditional ceramic preparation process, Bi_2_O_3_ (Chenguang Chemical Co., Ltd., Qishan, China, AR, 99%), TiO_2_ (Zhongxing Electronic Materials Co., Ltd., Xiantao, China, AR, 99.5%), Na_2_CO_3_ (Xilong Chemical Co., Ltd., Shantou, China, AR, 99.7%), K_2_CO_3_ (Xilong Chemical Co., Ltd., Shantou, China, AR, 99.8%), and Ag_2_O (Chenguang Chemical Co., Ltd., Qishan, China, AR, 99%) were used as starting materials. Based on the stoichiometric ratio of components, the system Bi_0.5_(Na_0.825_K_0.175_)_0.5_TiO_3_+x mol% Ag_2_O (x = 0, 0.1, 0.3, 0.5, 0.7, 0.9, abbreviated as BNKT-xA) was weighed. The raw material was ball milled for 6~7 h, fully mixed and crushed, and presynthesized at 850~900 °C for 2 h. The synthesized ceramic powders were fully ground and added with the proper amount of binder PVA, and then granulated to obtain particles with good fluidity. The green sheets with thickness of 1.2~1.5 mm and diameter of 10 mm were obtained by dry pressing under a certain pressure, and then sintered at 1150~1200 °C for 2 h to obtain ceramics. Under the silicone oil at 80 °C, the piezoceramics were poled by a DC electric field of 4~5 kV/mm for 20~30 min, and then the electrical properties of the samples were measured after one day and night. 

The phase structure of the piezoceramics was identified by XRD (Rigaku Dmax/RB, Tokyo, Japan). The surface of the piezoceramics was obtained by SEM (JSM-6510, JEOL, Akishima, Japan). The density of the ceramics was determined using the Archimedes method. The *d*_33_ of poled ceramic samples was measured by a Burlincourt-type *d*_33_ meter (ZJ-3A, Institute of Acoustic Academia Sinica, Beijing, China). The temperature-dependent dielectric properties were measured using an impedance analyzer (Agilent4284A, Santa Clara, CA, USA) and a custom-designed furnace. The measurements were conducted in the range of 25–500 °C with a heating rate of 2 °C/min. The *k_p_* was determined using Onoe’s formula, which involved analyzing the frequencies obtained from antiresonance and resonance measurements [25].

## 3. Results and Discussion

### 3.1. Structure

The XRD patterns of the BNKT-xA piezoceramics are presented in Figure 1. In Figure 1a, it is evident that all the structures of the BNKT-xA ceramics (x = 0–0.9%) show a single ABO_3_-typed perovskite, indicating the substitution of Ag ions for A-site cations in the Bi_0.5_(K, Na)_0.5_TiO_3_ lattices to form a homologous compound. The BNT-BKT piezoceramics are known to have a MPB (x = 0.16–0.20) between the R and T phases. In order to investigate the crystal structure of the BNKT-xA piezoceramics, magnified XRD patterns within the 2ɵ range of 39–47° are given in Figure 1b. In Figure 1b, the miller index is used to check special peaks because of the R–T phase boundary. The peaks at 46° correspond to the rhombohedral (024) and tetragonal (002) phases, based on the *R3c* and *P4bm* space group (COD ID: 2103295; COD ID: 2102068) [26,27]. Hence, the observed peak splitting demonstrates that all the BNKT-xA piezoceramics are situated in the R–T MPB. Previous studies by Wu et al. demonstrated that for 0 ≤ y ≤ 0.075, there is no significant change in the phase structure of (0.94-y)Na_0.5_Bi_0.5_TiO_3_-yBi_0.5_Ag_0.5_TiO_3_-0.06BaTiO_3_ piezoceramics within the MPB range [22]. Similarly, Lin et al. observed that Ag^+^ substitution for A-site cations does not affect the phase transformation in Bi_0.5_(Na_0.75−y_Li_0.075_K_0.175_Ag_y_)_0.5_TiO_3_ piezoceramics (0.015 ≤ y ≤ 0.15), which falls within the MPB [23]. Additionally, Liao et al. found that lower Ag content has no influence on the R–T phase transition in the (K_0.175_Na_0.825−y_Ag_y_)_0.5_Bi_0.5_TiO_3_ system (y = 0.015–0.075) [24]. Consistent with these previous reports, our results show a similar trend. Furthermore, with the appropriate addition of Ag^+^, the diffraction peaks gradually shift to smaller angles. During the sintering process, the evaporation of A-site cations often results in the generation of oxygen vacancies in BNKT ceramics. However, the dopant of Ag ions effectively counteracts the loss of A-site cations, thereby reducing the occurrence of oxygen vacancies. It is well established that an abundance of oxygen vacancies can cause the contraction of unit cells and a movement of diffraction peaks towards higher angles [28,29,30,31]. Consequently, the observed decrease in the displacement of the (002) peaks at low angles in the BNKT-xA ceramics demonstrates a reduction in the presence of oxygen vacancies.

Figure 2 illustrates the surface morphology of BNKT-xA piezoceramics using SEM. It is evident that the ceramics exhibit distinct grain boundaries and a regular square shape. With the increase in Ag^+^ content, the tetragonal feature of the grains is further strengthened, resulting in a more uniform and homogeneous shape. Therefore, the appropriate addition of Ag^+^ ions contributes to the homogeneity of the ceramic grains. This finding is consistent with a previous study on Bi_0.5_(Na_0.825−y_K_0.175_Ag_y_)_0.5_TiO_3_ ceramics [24]. Furthermore, the relative density of the piezoceramics goes up after the addition of Ag^+^. All BNKT-xA ceramics have high densities ranging from 5.62–5.71 g/cm^3^, which exceeds 95% of the theoretical values of Bi_0.5_(K_0.2_Na_0.8_)_0.5_TiO_3_ (5.889 g/cm^3^), as seen in Figure 3 [32]. The appropriate doping of Ag^+^ has enhanced the sinterability and densification of the BNKT-xA piezoceramics. This observation is likely due to the reduction in the evaporation of Na and K cations caused by the proper addition of Ag^+^. The homogeneous structure and dense microstructure are particularly advantageous for enhancing the electric properties of BNKT-xA piezoceramics.

### 3.2. Electrical Properties

Figure 4 illustrates the piezoelectric and dielectric properties of BNKT-xA piezoceramics. As the amount of Ag^+^ increases, both *d*_33_ and *k_p_* gradually go up as well. When x = 0.3%, *d*_33_ gets to its peak value at 147 pC/N, the electromechanical coupling coefficient *k_p_* peaks at 0.31 for x = 0.5%. The incorporation of Ag^+^ enhances the piezoelectric properties of BNKT-xA ceramics compared to pure BNKT ceramics. This enhancement can be attributed to a mechanism in which the sintering process leads to the partial volatilization of sodium and potassium, creating vacancies. Ag ions occupy these vacancies within the crystal lattice, causing distortion in the perovskite structure. This distortion is helpful to the migration of domain walls, promoting the reversal of electric domains [33]. Additionally, the introduction of an appropriate amount of Ag ions results in denser ceramics, as depicted in Figure 2 and Figure 3. The increased density allows for sufficient polarization, thereby further improving the piezoelectric properties of the BNKT-xA ceramics. Furthermore, Figure 4b demonstrates that similar to the correlation between x and *d*_33_, the relative permittivity attains its peak value (1199) at x = 0.3%, subsequently diminishing as the concentration of Ag^+^ increases. The dielectric loss obtains its minimum value of 0.063 at x = 0.3% before gradually increasing.

### 3.3. Dielectric Properties

Figure 5a illustrates the variations of *ε_r_* and tanδ in BNKT-xA piezoceramics with respect to temperature and frequency. Similar to BNT-based piezoceramics, there are two phase transitions observed within the measured temperature range: a transition from a ferroelectric to an anti-ferroelectric phase transformation (at *T_f_*), and a transition from an anti-ferroelectric to a paraelectric phase transformation (at *T*_m_) [19,34,35]. A single peak at *T_f_* is observed in the tan*δ*-*T* curves of the ceramics, and tanδ goes up significantly at temperatures higher than *T*_m_ due to substantial leakage conduction [36,37]. The variations of *ε_r_* reveal that the temperature-dependent behavior of the piezoceramics reveals an increase in *ε*_r_ as the temperature rises. This is characterized by a broadened peak around *T_f_* and the attainment of the maximum value at *T*_m_. In addition, with the addition of Ag^+^, a slight decrease in *T_f_* and *T*_m_ is observed among the different samples, as shown in Figure 5b. *T*_m_ serves as an indicator of the equilibrium of the oxygen octahedron, and it is able to be affected by the presence of ions [38]. Previous studies have demonstrated that the Ag-O bond has an energy of 221 kJ/mol, which is comparatively lower than the energy of K-O and Na-O bonds (271.5 kJ/mol, 270 kJ/mol) [39,40]. Hence, by properly adding Ag^+^, Ag ions occupy the A-site in the ABO_3_-typed lattice after the partial evaporation of alkali metal ions, thereby increasing the instability of oxygen octahedron and reducing *T_m_* [39,41].

In addition, the *ε*_r_-*T* curves show prominent characteristics of the diffuse phase transition. Specifically, with an increase in frequency, the *ε*_r_ exhibits a corresponding decrease, resembling the behavior of relaxor-like ferroelectric materials such as BNT-BT [10] and BNT-BKT [34]. Based on the aforementioned findings, it is evident that the temperature-dependent variation in the relative permittivity of BNKT-xA ceramics does not conform to the classical Curie–Weiss law when the temperature exceeds *T*_m_. In order to describe the diffusion of the phase transformation in the ferroelectric materials, a modified Curie–Weiss law is employed in our study [42].

The equation is as follows: (1)1εr−1εr,m=C(T−Tm)α

Here, *ε*_r_ and *ε*_r,m_ represent the dielectric constants at temperature *T* and *T*_m_, respectively. The diffusion constant (γ) is assumed to range between 1 and 2. If γ equals 1 or 2, the materials are referred to as normal or “complete” diffuse ferroelectrics. In contrast, when γ values fall within the range of 1 to 2, the materials are classified as relaxor ferroelectrics. Figure 6 illustrates the diffusion for BNKT-xA piezoceramics at 1 kHz. Through linear regression analysis of the experimental data, the slope of fitting curves is determined to identify the value of γ. In this case, the value of γ ranges between 1.90 and 1.96, demonstrating that the BNKT-xA piezoceramics exhibit a more diffuse behavior as the concentration of Ag^+^ increases. It can be reasonably ascribed to disorder among A-site cations caused by the presence of Ag^+^. Based on Shannon’s ionic radius, the radius of Ag^+^ is 1.26 Å. It is similar to that of K^+^ (1.33 Å) and Na^+^ (0.98 Å) [29]. Thus, Ag^+^ can occupy the A-site vacancies in the ABO_3_-typed structure after part of Na and K evaporates during the sintering procedure. Consequently, the presence of Ag^+^ induces disorder at the A-site cations, resulting in the observed diffuse phase transition characteristics. Moreover, Table 1 presents the diffusion exponent γ for BNKT-xA ceramics with x = 0–0.9% at 1 kHz, which exhibit a consistent linear relationship. Notably, the diffusion exponent γ exhibits a clear dependence on frequency, exhibiting higher values at higher frequencies, similar to the BNKT–x%MnCO_3_ ceramics [43]. Our findings, in line with those of other researchers, demonstrate that the introduction of Ag leads to an increase in the diffusion exponent.

### 3.4. Thermal Instability

The investigation of the depolarization temperature (*T*_d_) in BNT-based ceramics is of utmost significance in the field of device applications. In this study, *T*_d_ was determined by analyzing the temperature-dependent characteristics of piezoelectric properties. Figure 7 illustrates the relationship between *d*_33_ and temperature for the ceramics. Initially, *d*_33_ exhibits a slight decrease as the temperature increases up to 140 °C (*d*_33_ = 45 pC/N), followed by a significant decrease with further temperature increase. The disappearance of piezoelectric responses is observed within the temperature range of 150–160 °C. This indicates that the *T*_d_ of the piezoceramics is above 140 °C. The compound (Li_0.075_Na_0.925−x−y_Ag_y_K_x_)_0.5_Bi_0.5_TiO_3_, discovered by D.M. Lin et al. [23], exhibits a relatively high performance (*d*_33_ = 203 pC/N, *k*_p_ = 37.1%). This can be attributed to the partial substitution of Na ions in the Na_0.5_Bi_0.5_TiO_3_ lattice by K^+^ and Ag^+^, which facilitates the rotation of ferroelectric domains and improves the electrical performance of ceramics. However, it has a low *T*_d_ of approximately 98 °C, as observed in (Li_0.075_Na_0.675_Ag_0.075_K_0.175_)_0.5_Bi_0.5_TiO_3_ ceramics. Ag-doped Bi_0.5_(Na_0.825_K_0.175_)_0.5_TiO_3_ ceramics exhibit relatively low properties compared to (Li_0.075_Na_0.675_Ag_0.075_K_0.175_)_0.5_Bi_0.5_TiO_3_, but they demonstrate a high *T*_d_ (above 140 °C, Figure 7). The elevated *T*_d_ has a crucial role in device applications. It can be ascribed to the entry of Ag ions in A-site vacancies of Bi_0.5_(Na_0.825_K_0.175_)_0.5_TiO_3_ piezoceramics after the evaporation of partial Na^+^ and K^+^, promoting the coupling effect between A-site cations and TiO_6_, thereby increasing the depolarization temperature [19,44].

In addition, within the temperature range of 25 to 125 °C, the *d*_33_ of BNKT-xA ceramics remains in the range of 127–147 pC/N (change in *d*_33_ ≤ 13.6%). This behavior can be attributed to an underlying fact. It is suggested that the BNKT-xA piezoceramics undergo a structural transition from the R phase to the T phase within the temperature range of 25 to 125 °C, enhancing their thermal stability. However, as the temperature continues to rise, the ceramics progressively move away from the ferroelectric phase and transition towards the anti-ferroelectric phase, as depicted in Figure 5. This progressive shift ultimately results in a decrease in *d*_33_.

## 4. Conclusions

The present study systematically investigates the effect of Ag^+^ on the crystal structure, microstructure, and electrical performance of (K_0.175_Na_0.825_)_0.5_Bi_0.5_TiO_3_ Pb-free ceramics, which were prepared using sintering techniques. The BNKT-xA lead-free ceramics exhibit characteristic diffraction peaks associated with pure perovskites. Furthermore, the compositions demonstrate good sintering properties, resulting in a dense microstructure. The incorporation of Ag ions slightly affects the *T*_m_ of BNKT-xA piezoceramics by destabilizing the stability of the oxygen octahedron. The observed relaxor behavior in the ceramics is ascribed to the disorder of cations occurring at the A-sites. Simultaneously, the investigated piezoceramics have good electrical properties, characterized by a piezoelectric constant (*d*_33_) value of 147 pC/N and a coupling coefficient (*k*_p_) of 29.6%. The *d*_33_ value of the BNKT-xA piezoceramics demonstrates consistent stability within a narrow range of 127–147 pC/N (with a maximum variation of *d*_33_ ≤ 13.6%) over a wide temperature range spanning from 25 to 125 °C. This finding indicates the reliable performance of the piezoceramics across various operational temperatures.

## Figures and Tables

**Figure 1 materials-16-05342-f001:**
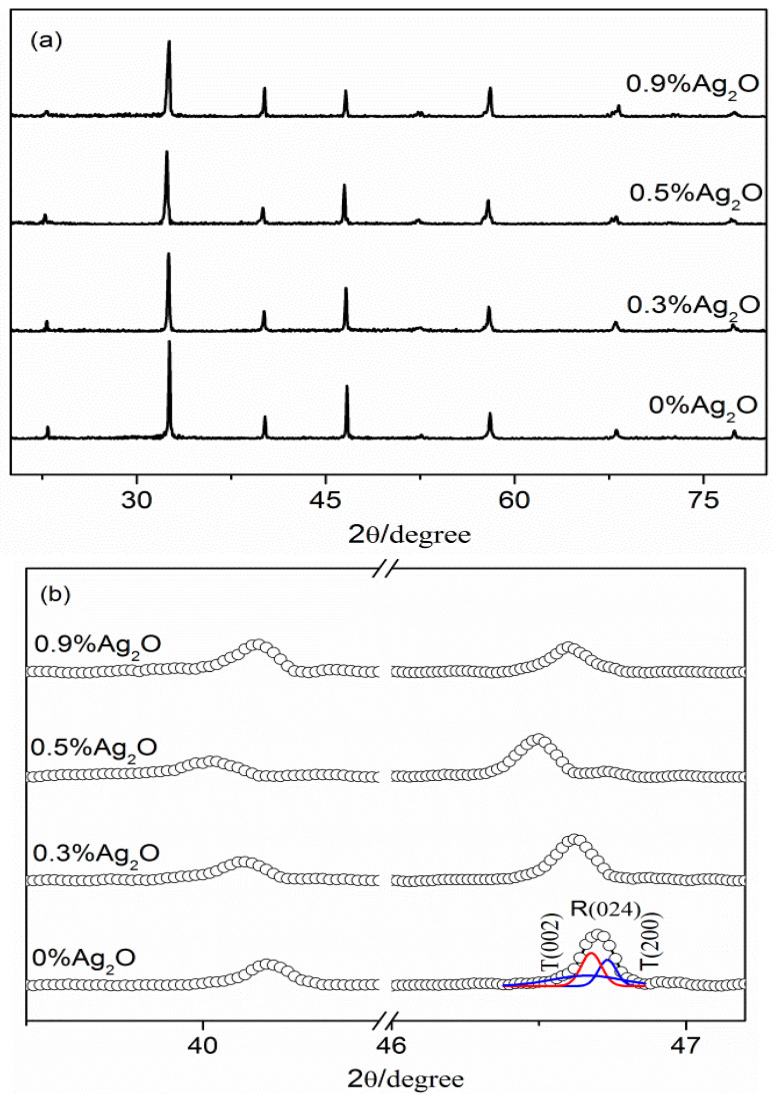
(**a**) XRD patterns of BNKT-xA piezoceramics; (**b**) the expanded XRD patterns of BNKT-xA piezoceramics in the range of 35–47°.

**Figure 2 materials-16-05342-f002:**
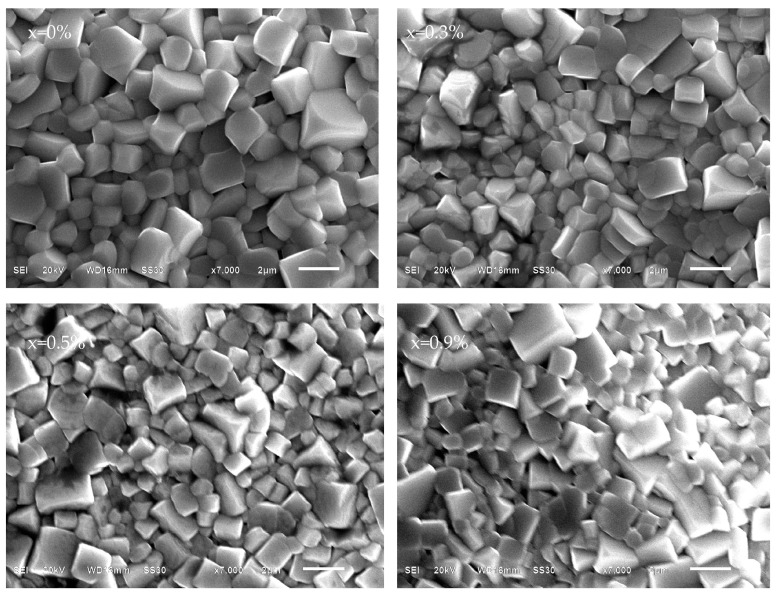
SEM surface micrographs of BNKT-xA ceramics.

**Figure 3 materials-16-05342-f003:**
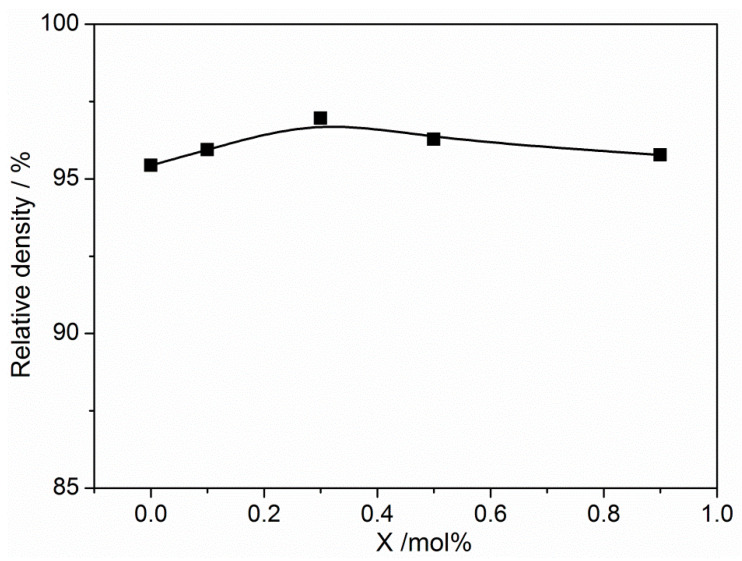
Relative density of BNKT-xA ceramics.

**Figure 4 materials-16-05342-f004:**
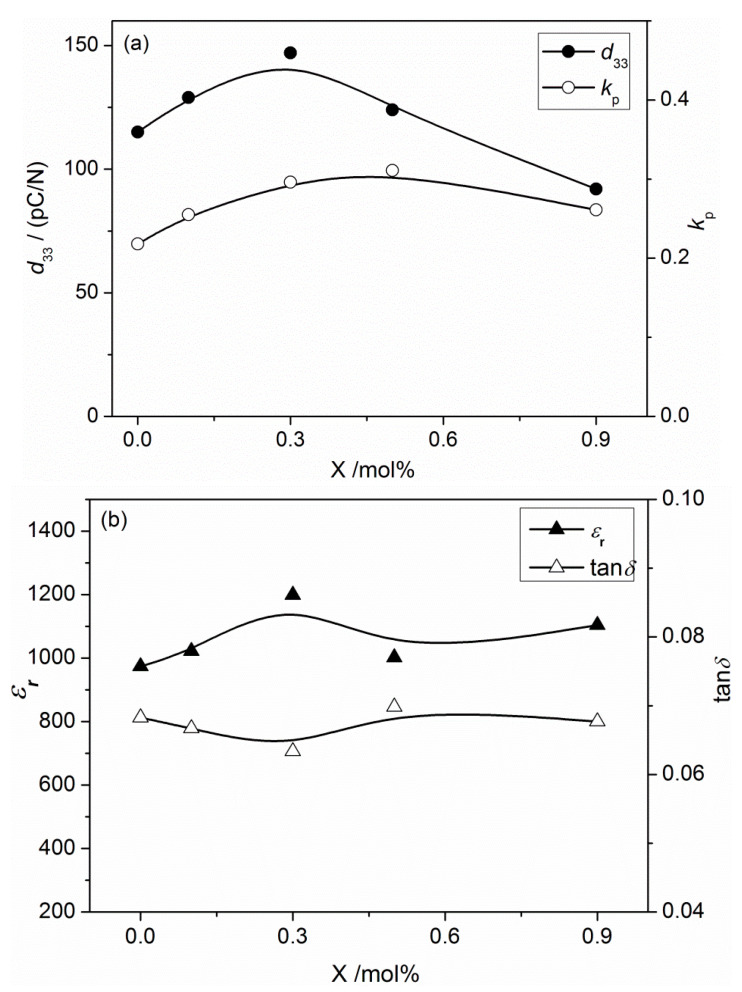
(**a**) Piezoelectric and (**b**) dielectric properties of BNKT-xA ceramics measured at 1 kHz.

**Figure 5 materials-16-05342-f005:**
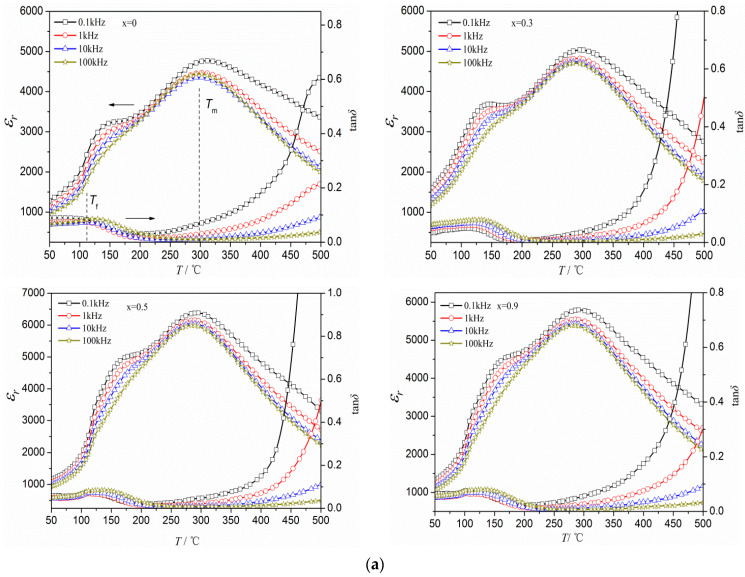
(**a**) Temperature dependence of dielectric properties for BNKT-xA piezoceramics. (**b**) *T_f_* and *T*_m_ for BNKT-xA piezoceramics measured at 1 kHz.

**Figure 6 materials-16-05342-f006:**
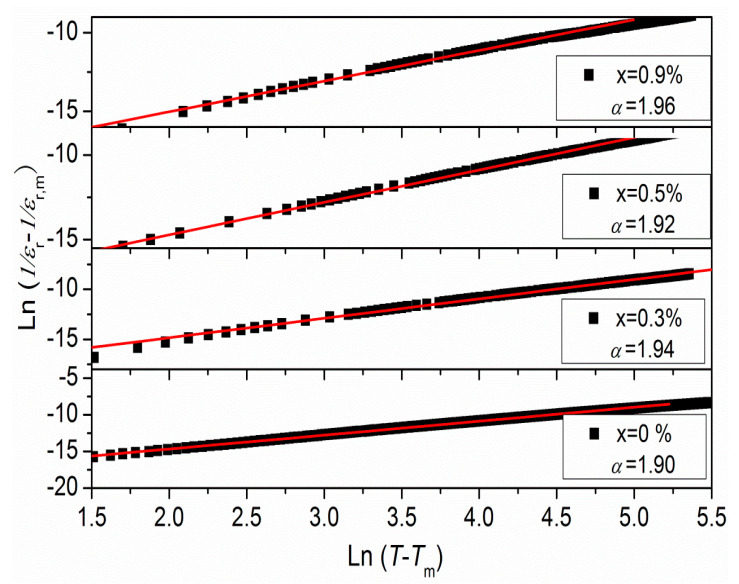
ln(1/ε_r_ − 1/ε_r,m_) vs. ln(T − T_m_) for the BNKT-xA piezoceramics (1 kHz).

**Figure 7 materials-16-05342-f007:**
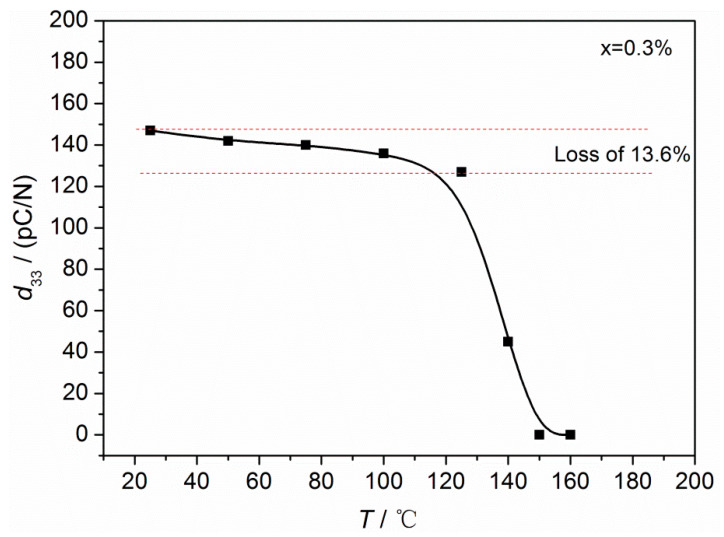
Temperature dependence of *d*_33_ for BNKT-xA ceramics.

**Table 1 materials-16-05342-t001:** The diffusion constants and electrical properties of BNKT-xA lead-free ceramics measured at 1 kHz.

x/%	γ	*T_f_*/°C	*T*_m_/°C	*d*_33_*/*(pC/N)	*k_p_*	*ε_r_*	tan*δ*
0	1.902	114	300	115	0.218	974	0.06827
0.3	1.947	109	296	147	0.296	1199	0.06337
0.5	1.919	111	293	124	0.311	1002	0.06983
0.9	1.957	105	291	92	0.261	1104	0.0677

## Data Availability

Data is available from the corresponding author upon reasonable request.

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
