# Peer review of "Piezoelectric and Dielectric Properties in Bi0.5(Na,K)0.5TiO3-x Ag2O Lead-Free Piezoceramics"

_materials, 2023, doi:10.3390/ma16155342_

Round 1
Reviewer 1 Report
The manuscript was written well. The Data presented were not sufficiently scientifically discussed. If the authors can relate their results with other relevant works and concepts, it will improve the quality of the manuscript for better readership
According to the traditional ceramic preparation process, Bi2O3 (99%), TiO2 (99.5%), 77 Na2CO3 (99.7%), K2CO3 (99.8%) and Ag2O (99%) were used as starting materials, and the 78 system Bi0.5(Na0.825K0.175)0.5TiO3+ x mol % Ag2O (x=0, 0.1, 0.3, 0.5, 0.7, 0.9 , abbreviated as 79 BNKT xA) was weighed according to the stoichiometric ratio of components.
It starts and ends with "according".. reframe the sentence and do similar checks throughout the manuscript
Author Response
Response to Reviewer 1 Comments
Point 1: The manuscript was written well. The Data presented were not sufficiently scientifically discussed. If the authors can relate their results with other relevant works and concepts, it will improve the quality of the manuscript for better readership。
Response 1: Thanks for your comment. The relationship between dielectric properties and relaxors have been systematically studied again. Table 1 is given. A comprehensive elucidation of the underlying mechanism is presented in detail at the Paragraph4, Section3.
Point 2: According to the traditional ceramic preparation process, Bi2O3 (99%), TiO2 (99.5%), 77 Na2CO3 (99.7%), K2CO3 (99.8%) and Ag2O (99%) were used as starting materials, and the 78 system Bi0.5(Na0.825K0.175)0.5TiO3+ x mol % Ag2O (x=0, 0.1, 0.3, 0.5, 0.7, 0.9 , abbreviated as 79 BNKT xA) was weighed according to the stoichiometric ratio of components.
It starts and ends with "according".. reframe the sentence and do similar checks throughout the manuscript
Response 2: Thanks for your comment. The sentence is changed as “According to the traditional ceramic preparation process, Bi2O3 (99%), TiO2 (99.5%), Na2CO3 (99.7%), K2CO3 (99.8%) and Ag2O (99%) were used as starting materials. Based on the stoichiometric ratio of components, the system Bi0.5(Na0.825K0.175)0.5TiO3+ x mol % Ag2O (x=0, 0.1, 0.3, 0.5, 0.7, 0.9 , abbreviated as BNKT-xA) was weighed.” Besides, the related sentences have been checked and revised.

Reviewer 2 Report
This manuscript presents the synthesis of perovskite phase Bi0.5(Na,K)0.5TiO3 modified with AgO and discusses its detailed electrical properties. The sample is well characterized, and the data is explained well. However, some following issues should be addressed before accepting in Materials.
1. In the Introduction section, the author stated that some main groups of lead-free piezoelectric ceramics, however, did not include the references. Some previous works should be added, it is also possible to add to this work as bi-layered phase and niobate-phase ceramics: https://doi.org/10.1016/j.ceramint.2022.01.307 ; https://doi.org/10.1016/j.jeurceramsoc.2020.09.032.
2. The author presents the formula for Bi0.5(Na,K)0.5TiO3-xAgO2 in this work, does the addition of x mol % AgOmeans that Ag ions substitute the A cation in the pristine NBT phase, or is this AgO only added with a fixed NBT composition? Does this formula mean Bi0.5(Na1-x-yKxAgy)0.5TiO3, where Ag substitutes another cation in the A position of perovskite? If yes, should the author write this formula as 1-xBi0.5(Na,K)0.5TiO3-xAgO2?
3. The sub-section of 3.1 should be Crystal Structure, and the SEM discussion should be written as a different sub-section 3.2. Morphology observation.
4. In the XRD spectrum (Fig. 1(a)), the author should add the miller index of each peak to further confirm that the phase formed is a single-phase perovskite.
5. In the electrical properties of Fig. 4, the author should add the measurement conditions such as temperature, frequency, or others (in Figure or caption), since it affects the electrical value. The caption of Figure 4 should also be made clearer by the differences between a and b.
6. I suggest the author list all values of electrical properties (εr, tanδ, Tf, Tm, γ, d33, etc) for all compositions in a Table; it is easier to compare all values with different x compositions. I suggest the comparison of all values at 10 kHz (or higher), since in the high-frequency regime the contribution of external factors on electrical properties is reduced. Therefore, the author should recalculate the γ value in Figure 6 for the 10 kHz, not 1 kHz.
7. Some minor corrections in the manuscript:
Adding the trademark of all oxides and chemicals used in the material section.
All symbols of A and B site cations should be in italics.
Captions of Figures 6 and 7 should not be in italics.
Author Response
Response to Reviewer 2 Comments
Point 1: In the Introduction section, the author stated that some main groups of lead-free piezoelectric ceramics, however, did not include the references. Some previous works should be added, it is also possible to add to this work as bi-layered phase and niobate-phase ceramics: https://doi.org/10.1016/j.ceramint.2022.01.307 ; https://doi.org/10.1016/j.jeurceramsoc.2020.09.032.
Response 1: Thanks for your comment. It has been added at the relate part (Ref 5 and 6).
Point 2: The author presents the formula for Bi0.5(Na,K)0.5TiO3-xAgO2 in this work, does the addition of x mol % AgOmeans that Ag ions substitute the A cation in the pristine NBT phase, or is this AgO only added with a fixed NBT composition? Does this formula mean Bi0.5(Na1-x-yKxAgy)0.5TiO3, where Ag substitutes another cation in the A position of perovskite? If yes, should the author write this formula as 1-xBi0.5(Na,K)0.5TiO3-xAgO2?
Response 2: a fixed BNKT composition is only doped by AgO. Hence ,we had used Bi0.5(Na,K)0.5TiO3-xAg2O to express the solutions.
Point 3: The sub-section of 3.1 should be Crystal Structure, and the SEM discussion should be written as a different sub-section 3.2. Morphology observation.
Response 3: Thanks for your comment. Crystal Structure and Morphology observation belong to the part of structure.
Point 4: In the XRD spectrum (Fig. 1(a)), the author should add the miller index of each peak to further confirm that the phase formed is a single-phase perovskite.
Response 4: Thanks for your comment.Compared to pure BNKT ceramics, Ag-doped BNKT ceramics have the same crystal structure, hence, a single ABO3-typed perovskite is kept. In Fig1(b),the miller index is used to check special peaks because of R-T phase boundary.
Point 5: In the electrical properties of Fig. 4, the author should add the measurement conditions such as temperature, frequency, or others (in Figure or caption), since it affects the electrical value. The caption of Figure 4 should also be made clearer by the differences between a and b.
Response 5: The measurement conditions have been added in caption of Fig. 4.
The caption of Figure 4 has been changed as” Figure 4 (a) piezoelectric and (b)dielectric properties of BNKT-xA ceramics meas-ured at 1kHz”
Point 6: I suggest the author list all values of electrical properties (εr, tanδ, Tf, Tm, γ, d33, etc) for all compositions in a Table; it is easier to compare all values with different x compositions. I suggest the comparison of all values at 10 kHz (or higher), since in the high-frequency regime the contribution of external factors on electrical properties is reduced. Therefore, the author should recalculate the γ value in Figure 6 for the 10 kHz, not 1 kHz.
Response 6: Thanks for your comment. Table 1 is given at the end of Paragraph6, Section3,which show the diffusion constants of BNKT-xA lead-free ceramics at various frequencies
Point 7: Some minor corrections in the manuscript:
Adding the trademark of all oxides and chemicals used in the material section.
All symbols of A and B site cations should be in italics.
Captions of Figures 6 and 7 should not be in italics.
Response 7: The trademark of all used oxides and chemicals has been added in the material section.
And the related part has been revised.

Reviewer 3 Report
The authors report a study of the piezoelectric properties of polycrystalline Bi0.5(Na0.825K0.175)0.5TiO3 doped with varying amounts of Ag2O. The study is well-motivated in the introduction and the experimental results are generally sound and convincing. The paper may be published, once the authors have addressed the following concerns:
1. The XRD patterns of the synthesized ceramics indicate them to be single-phase, with the cell volume increasing upon increasing amount of Ag2O. However, this latter trend does not hold for the highest amount of Ag2O, 0.9%. Similarly, the increase of the piezoelectric coefficient d33, observed for lower Ag2O concentrations, is not present for 0.9% Ag2O. It should be discussed why addition of small amounts of Ag2O works, but addition of larger amounts does not.
2. It would be good to verify the actual chemical composition of the as-synthesized ceramics e.g. by EDX.
3. The temperature-dependence of d33 of the ceramics synthesized with 0.3% Ag2O is shown in Fig. 7, demonstrating small variation up to 130 C. It would be instructive to show this temperature dependence also for the other compositions. At least the corresponding curve for 0% Ag2O should be shown for comparison.
4. The conclusions state that Ag-addition significantly affects the transition temperature (Tm) – this is not completely obvious from Fig. 5 and could be worked out in more detail. Furthermore, with respect to the piezoelectric properties, the effect of Ag-addition on Tf would be of even higher interest.
Language quality is largely ok.
Author Response
Response to Reviewer 3 Comments
Point 1: The XRD patterns of the synthesized ceramics indicate them to be single-phase, with the cell volume increasing upon increasing amount of Ag2O. However, this latter trend does not hold for the highest amount of Ag2O, 0.9%. Similarly, the increase of the piezoelectric coefficient d33, observed for lower Ag2O concentrations, is not present for 0.9% Ag2O. It should be discussed why addition of small amounts of Ag2O works, but addition of larger amounts does not.
Response 1: Thanks for your comment. The related mechanism is as follows: The incorporation of Ag+ enhances the piezoelectric properties of BNKT-xA ceramics compared to pure BNKT ceramics. This enhancement can be attributed to a mechanism in which the sintering process leads to the partial volatilization of sodium and potassium, creating vacancies. Ag ions occupy these vacancies within the crystal lattice, causing distortion in the perovskite structure. This distortion is helpful to the migration of domain walls, promoting the reversal of electric domains. Additionally, the introduction of an appropriate amount of Ag ions results in denser ceramics, as depicted in Figures 2 and 3. The increased density allows for sufficient polarization, thereby further improving the piezoelectric properties of BNKT-xA ceramics.( Paragraph3, Section3)
Point 2: It would be good to verify the actual chemical composition of the as-synthesized ceramics e.g. by EDX.
Response 2: Thanks for your advice. Recently, the author's unit is moving to a new campus, SEM has been dismantled, and it is preparing to move to the new campus, therefore, temporarily unable to carry out EDS test. Nevertheless, the corresponding testing will be carried out in the future.
Point 3: The temperature-dependence of d33 of the ceramics synthesized with 0.3% Ag2O is shown in Fig. 7, demonstrating small variation up to 130 C. It would be instructive to show this temperature dependence also for the other compositions. At least the corresponding curve for 0% Ag2O should be shown for comparison.
Response 3: Thanks for your advice. BNKT-0.3mol%A piezoceramics show remarkable improvements in their properties(d33=147pC/N, kp=29.6%,ε=1199, tanδ=0.063). Therefore, Td was determined by analyzing the temperature-dependent characteristics of piezoelectric properties of BNKT- 0.3mol%A piezoceramics. For pure BNKT , the Td had been measured in the our previous work, you can find it at the following reference:
Chen Xiaoming,Liao Yunwen,Mao Lijun, et al. Microstructure and piezoelectric properties of Li-doped Bi0.5(Na0.825K0.175)0.5TiO3 piezoelectric ceramics [J]. Phys Status Solidi A, 2009, 206(7): 1616-9.
Point 4: The conclusions state that Ag-addition significantly affects the transition temperature (Tm) – this is not completely obvious from Fig. 5 and could be worked out in more detail. Furthermore, with respect to the piezoelectric properties, the effect of Ag-addition on Tf would be of even higher interest.
Response 4: Thanks for your comment. I made a mistake. The related sentence is changes as follow: The incorporation of Ag ions slightly affects the Tm of BNKT-xA piezoceramics by destabilizing the stability of the oxygen octahedron. In addition,the investigation of the depolarization temperature (Td) in BNT-based ceramics is of utmost significance in the field of device applications. Therefore , Td is checked and the related mechanism is given in Paragraph 7-8, Section3.

Reviewer 4 Report
Report on the manuscript
Title: Piezoelectric and dielectric properties in Bi0.5(Na,K)0.5TiO3-x 2 Ag2O lead-free piezoceramics
Authors: Xiaoming Chen and Yunwen Liao
Manuscript Number: materials-2509262
In my opinion, this manuscript's contents significantly contribute to the field of the dielectric properties of piezoelectric materials.
The authors obtained important and argued results.
However, before the Editor decides, I suggest that the authors must take into account the following corrections:
1. Abstract is short and should be modified.
2. Authors, please improve the quality of editing for equation (1) and use italics (Math Equation type and for math symbols.
3. Uniformize the References:
It is written the name of the journal with or without abbreviations; please choose the form with abbreviations and put the DOI where it is possible.
4. I think the authors need to emphasize more clearly the contribution of the manuscript from a scientific point of view.
5. References are suggestive. I am convinced that it is useful for the manuscript if it will be included in the References section more recent papers with the same topics, or using similar procedures, for example:
- Dielectric and Electrical Properties of BLT Ceramics Modified by Fe Ions, Materials 2020, 13(24), 5623, DOI:10.3390/ma13245623;
- Cracks interaction in a pre-stressed and pre-polarized piezoelectric material, Journal of Mechanics, 36, 177-182, (2020), DOI: 10.1017/jmech.2019.57.
If the authors consider these corrections, then, without any doubts, this manuscript deserves to be published.
Minor editing of English language required.
Author Response
Response to Reviewer 4 Comments
Point 1: Abstract is short and should be modified.
Response 1: Thanks for your comment. Abstract is modified.
Point 2: Authors, please improve the quality of editing for equation (1) and use italics (Math Equation type and for math symbols.
Response 2: Thanks for your comment. It is modified.
Point 3: Uniformize the References:
It is written the name of the journal with or without abbreviations; please choose the form with abbreviations and put the DOI where it is possible.
Response 3: Thanks for your comment. It is modified by use of Endnote software.
Point 4: I think the authors need to emphasize more clearly the contribution of the manuscript from a scientific point of view.
Response 4: Thanks for your comment. In this case, an appropriate amount of Ag2O was introduced into BNT-Bi0.5K0.5TiO3 to enhance its sinterability and performance. The traditional ceramic process is em-ployed to prepare the piezoceramics.The structure and electrical properties are systematically studied. A comprehensive elucidation of the underlying mechanism is presented in detail.
Point 5: References are suggestive. I am convinced that it is useful for the manuscript if it will be included in the References section more recent papers with the same topics, or using similar procedures, for example:
- Dielectric and Electrical Properties of BLT Ceramics Modified by Fe Ions, Materials 2020, 13(24), 5623, DOI:10.3390/ma13245623;
- Cracks interaction in a pre-stressed and pre-polarized piezoelectric material, Journal of Mechanics, 36, 177-182, (2020), DOI: 10.1017/jmech.2019.57.
Response 5: Thanks for your comment. It has been added at the relate part (Ref 3 and 4)

Round 2
Reviewer 2 Report
I am satisfied that the authors revise the manuscript as per the reviewer's suggestions and explain the revision clearly in the response letter, this work can be accepted for publication in Materials after the minor revision is made.
1. I asked the author previously to list the electrical properties. Maybe the author misunderstood my suggestion, therefore, I suggest revising Table 1 to list all values of electrical properties (εr, tanδ, Tf, Tm, γ, d33, etc) for all compositions with the same frequency (1 kHz) not γ values at all frequency. From this table, the reader can know the magnitude of electrical value and compare changes in values more easily with variations in the composition of x.
Author Response
Response to Reviewer 2 Comments
Point 1: I am satisfied that the authors revise the manuscript as per the reviewer's suggestions and explain the revision clearly in the response letter, this work can be accepted for publication in Materials after the minor revision is made.
I asked the author previously to list the electrical properties. Maybe the author misunderstood my suggestion, therefore, I suggest revising Table 1 to list all values of electrical properties (εr, tanδ, Tf, Tm, γ, d33, etc) for all compositions with the same frequency (1 kHz) not γ values at all frequency. From this table, the reader can know the magnitude of electrical value and compare changes in values more easily with variations in the composition of x.
Response 1: Thanks for your comment. In Figure 4, piezoelectric and dielectric properties (εr, tanδ, d33, kp) of BNKT-xA ceramics are given ,which are measured at 1kHz. Besides, the Fig.5(b) has been added at the related part, which show Tf and Tm for BNKT-xA piezoceramics measured at 1kHz.
